# UNSUPERVISED DETECTION OF RECURRENT PATTERNS IN NEURAL RECORDINGS WITH CONSTRAINED FILTERS

## ABSTRACT

Spontaneous neural activity, characterized by the expression of repetitive patterns, is crucial for memory, learning and spatial navigation. However, further study of these patterns' functional role has been difficult due to a lack of scalable methods for their detection in large recordings. To address this challenge, we propose an unsupervised method which relies on backpropagation to optimize the parameters of a fixed number of spatiotemporal filters used as pattern detectors. We demonstrate the scalability and efficiency of our approach for detecting place cell sequences in biologically plausible synthetic and real datasets recorded from the mouse hippocampus. Our speed benchmarks show that our method significantly outperforms prior art, opening new possibilities for the analysis of spontaneous activity in larger recordings.

## 1 INTRODUCTION

A fundamental property of biological neural networks – and one that distinguishes them from the majority of modern deep neural networks – is the ability to change state not only in response to external input, but also spontaneously (Arieli et al., 1996; Beggs & Plenz, 2003). A prominent example of this is what is known as "hippocampal replay" (Lee & Wilson, 2002; Foster & Wilson, 2006; Pfeiffer & Foster, 2013), normally observed in animals during sleep or periods of immobility, which represents structured reactivation of neural activity patterns present during a behavioral task performed before. The importance of hippocampal replay has been shown to be crucial for memory, learning and navigation (Girardeau et al., 2009). In addition, animals' behavior in response to external stimuli depends on the structure of spontaneous activity before and at the time of stimulation (Fiser et al., 2004), which raises the intriguing possibility that spontaneous activity might encode sensory priors and therefore be a form of biological memory.

To address questions about the role of structured spontaneous activity, a number of methods have been proposed for unsupervised detection of neural activity patterns in the absence of an observable behavioral reference. These existing methods perform well and in reasonable time on modestly sized datasets. However, the study of spontaneous activity would benefit from the analysis of much larger datasets (with hundreds of neurons recorded over several days), which calls for more scalable pattern detection methods.

We introduce an efficient and scalable method for unsupervised detection of sequential patterns of neural activity based on optimizing a set of constrained spatiotemporal filters. Distinct from existing approaches (e.g. *convNMF*, *seqNMF*), we optimize the filters with backpropagation, which allows us to take advantage of popular automatic differentiation frameworks and GPU acceleration. To reduce the number of learnable parameters, we also propose an alternative formulation of our method, in which the filters themselves are parameterized as fixed-width truncated Gaussians. Our speed benchmarks show that the method, which we call *convSeq*, works significantly faster than existing pattern detection methods.

Our main contributions are as follows:

1. Our method advances the SOTA in terms of speed: given the same dataset, it performs over a 100 times faster than similar recently published methods;

2. Unlike *convNMF* and *seqNMF*, which are conceptually similar to our method, ours provides uncertainty estimates for the patterns detected, without requiring multiple optimization runs;

The rest of the paper is structured as follows. Section 2 offers a brief review of existing methods for detecting patterns in neural data. In Section 3 we introduce two formulations of our approach. In Section 4 we showcase its ability to detect various patterns in synthetic and real data, as well as accuracy and speed comparisons with a selection of other methods. We discuss the limitations and future directions in Section 5 and conclude with Section 6.

## 2 RELATED WORK

In general, classic methods working under linear assumptions, such as PCA and ICA (Jutten & Herault, 1991), struggle to capture spatio-temporal patterns in neural activity, as they tend to merge them into a single "large component" (Peter et al., 2016; Williams et al., 2020). This key limitation motivated many previous works which proposed alternative methods for detecting spatiotemporal structure in neural data, without using external reference events. For example, Watanabe et al. (2019) used edit similarity as a distance metric between potential spike patterns to identify cell assembly sequences. Quaglio et al. (2017) utilized conceptual stability to identify repeating spike patterns and Schrader et al. (2008); Torre et al. (2016) proposed using an "intersection matrix" to detect synchronous spike events (aka synfire chains), albeit in synthetic data. Shimazaki et al. (2012) proposed detecting higher-order spike correlations using state-space modeling. More recently, Grossberger et al. (2018) proposed a clustering method based on optimal transport, while Williams et al. (2020) designed a point process model of spike sequences utilizing a fully probabilistic Bayesian framework, and Stringer et al. (2023) proposed sorting neural responses along a one-dimensional manifold to expose the patterns.

The convolutional non-negative matrix factorization (*convNMF*) proposed by Smaragdis (2004; 2006) and first applied to in-vitro neural data by Peter et al. (2016) is conceptually closest to our approach. *convNMF* and its improved derivative, *seqNMF* (Mackevicius et al., 2019), aim to jointly estimate both the templates of the recurrent patterns and the time course of their activity. In contrast, our approach, as we describe next, only optimizes the templates (which we call "filters") and does so using backpropagation.

## 3 METHODS

The input to our model is a binary matrix $\mathbf{X} \in \{1,0\}^{N \times T}$, which represents a simultaneous recording of $N$ neurons for $T$ time bins (also referred to as "time steps"), such that $X_{n,t} = 1$ if there is a spike on the $n$-th neuron in time bin $t$, and $X_{n,t} = 0$ otherwise. We seek to find $K$ 2D filters $\mathbf{W}^{(k)} \in \mathbb{R}^{N \times M}$, such that each of them responds preferentially to one of $K$ unknown patterns defined here as repeating sequences of spikes. Each of the $K$ patterns repeat *inexactly* (due to variations in the relative timing (jitter) of spikes) an unknown number of times. The choice of $M$ and $K$ depends on the length (in time steps) and number of the patterns assumed to be present in the data.

### 3.1 FORMULATION WITH DIRECT FILTER OPTIMIZATION

We first describe how the filters $\mathbf{W}^{(k)}$, $k \in \{1, \ldots, K\}$, can be found by minimizing the following loss function:

$$\mathcal{L}(\mathbf{W}) = \sum_{k=1}^{K} -\text{Var}(\hat{\mathbf{x}}^{(k)}) + \beta_{\text{TV}} \text{TV}(\hat{\mathbf{x}}^{(k)}) + \beta_{\text{xcor}} \sum_{l>k}^{K} \rho_{\hat{\mathbf{x}}^{(l)} \hat{\mathbf{x}}^{(k)}}[j] \tag{1}$$

where $\hat{\mathbf{x}}^{(k)} = \text{softmax}(\mathbf{W}^{(k)}) * \mathbf{X}$, and "*" stands for convolution. The convolution is performed with zero padding only along the time dimension to ensure that $\hat{\mathbf{x}}^{(k)}$ has shape $1 \times T$. Softmax is computed over the time dimension of $\mathbf{W}^{(k)}$. $\text{TV}(\hat{\mathbf{x}}^{(k)}) = \frac{1}{T} \sum_{t=1}^{T-1} (\hat{x}_t^{(k)} - \hat{x}_{t+1}^{(k)})^2$ and $\sum_{l>k}^{K} \rho_{\hat{\mathbf{x}}^{(l)} \hat{\mathbf{x}}^{(k)}}[j]$ are total variation and cross-correlation over $j$ time steps, respectively. The first term in Eq. 1 maximizes the variance of the $k$-th filter's total response to the data. The idea is that if there exists

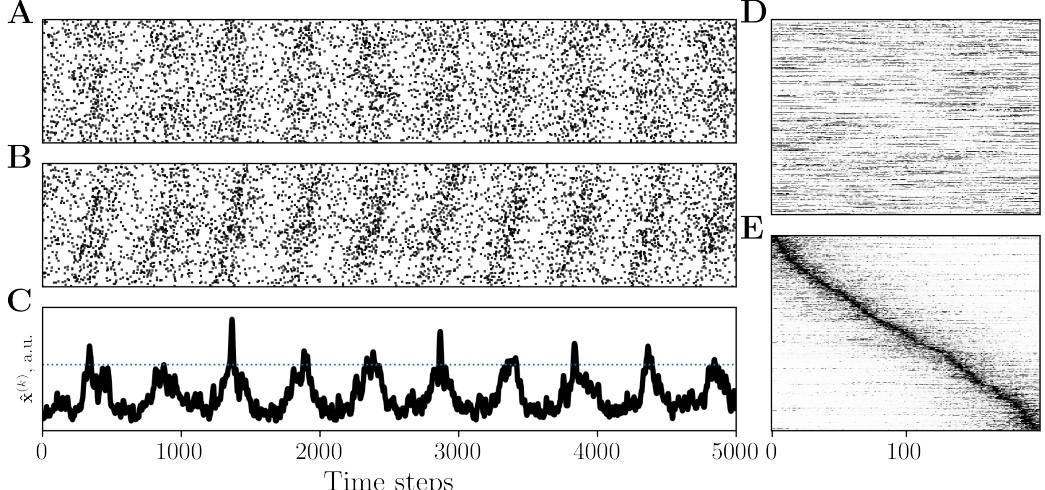

Figure 1: Original data matrix (A). The optimized filter (D) is sorted (E), and the sorting indices are used to rearrange the rows of the data matrix to expose the sequences (B). Peaks in $\hat{\mathbf{x}}^{(k)}$ exceeding the significance threshold $\alpha$ (dotted line) indicate significant detections of the pattern (C). In A, B, D, E the y-axis corresponds to neuron IDs.

a repeating pattern, the right filter (when convolved with the data) will produce peaks at the times of that pattern's occurrence. Importantly, while each filter's total response stays constant (that is $\sum \left[ \text{softmax}(\mathbf{W}^{(k)}) * \mathbf{X} \right] = \sum \mathbf{X}$), the variance of its total response is maximized when the filter has a good match with some repeating pattern. Keeping the filter's total response constrained makes it easy to bootstrap confidence intervals for the height of peaks in $\hat{\mathbf{x}}^{(k)}$, which can be used for testing the significance of the patterns detected (Sec. 3.4). The total variation term helps reduce the filters' response to background activity (i. e. neural activity unrelated to any pattern), reduce the false positive rate, and facilitate visual interpretation of results (Appendix E). Finally, the cross-correlation term in Eq. 1 encourages filter diversity when $K > 1$. That is, it prevents the filters from becoming "tuned" to the same (stronger or overrepresented) pattern. The weights of the total variation and cross-correlation penalty terms as well as other hyperparameters are listed in Appendix A.

## 3.2 Visualization of structured spontaneous activity

The presence, strength and temporal location of the patterns is captured by $\hat{\mathbf{x}}^{(k)}$: its peaks correspond to the times at which the pattern is expressed in neural activity (Fig. 1 C). These peaks alone, however, only suggest the presence of a pattern, and it is desirable to represent the data in a way that makes the detected structure clearly visible (e. g. in hippocampal recordings in which theta sequences are expected). To reveal the patterns, the optimized filters are sorted so that per-row maxima become temporally ordered. The sorting indices are used to rearrange the order of neurons in $\mathbf{X}$. We summarize this in Fig. 1 and Appendix C. We also note that depending on pattern complexity and strength, as well as parameter initialization, variations of the recovered patterns' shape are to be expected.

## 3.3 Formulation with parameterized Gaussian filters

In the above formulation, we seek to optimize the randomly initialized filters $\mathbf{W}^{(k)}$ directly, which means $N \times W \times K$ trainable parameters. However, assuming that patterns are sequences of spikes, whose relative timing is distorted by spike timing jitter, and that this jitter is Gaussian, we can reduce the number of trainable parameters by a factor of $N$. Specifically, at each optimization step, we can parameterize the $n$-th row in the $k$-th filter $\mathbf{W}^{(k)}$ as a truncated Gaussian function $f(\cdot)$ with mean $\mu_n^{(k)}$ and a fixed value of $\sigma$. In this way, we only need to optimize the means of the Gaussians in each row. In this formulation, the softmax function is no longer needed as the filter's impulse response is now constrained by the Gaussian function truncated to the filter's width $M$: $\hat{\mathbf{x}}^{(k)} = \mathbf{W}^{(k)} * \mathbf{X}$,

such that $\mathbf{W}_{n,:}^{(k)} = f(\mu_n^{(k)}, \sigma^2, 1, M)$, and $n \in \{1, \ldots, N\}$. While in terms of speed this formulation performs on par with the one described in Section 3.1, it offers a way to steer the model towards specific solutions by incorporating inductive biases into the filter design. For example, it should also be possible to learn per-neuron standard deviations $\sigma_n^{(k)}$ (although at the expense of doubling the number of trainable parameters) to capture each neuron's temporal jitter and its degree of participation in a pattern, but we leave this question to future work.

### 3.4 STATISTICAL TESTING

We consider the detection of the $k$-th pattern to be statistically significant at some time step $t$ if $\hat{x}_t^{(k)} >= \alpha$, where $t \in \{1, \ldots, T\}$ and $\alpha$ is a significance criterion, which is determined for each dataset individually. To estimate $\alpha$, we construct 1000 random filters and get $\hat{\mathbf{x}}_0^{(k)} = \mathbf{X} * \mathbf{W}_0^{(k)}$, $k \in \{1, \ldots, 1000\}$ . We construct the null distribution out of $\hat{\mathbf{x}}_0^{(k)}$, compute its mean, $\mu_0$, and standard deviation, $\sigma_0$ and set $\alpha$ to be four[1] standard deviations above the mean, i.e. $\alpha = 4\sigma_0 + \mu_0$ (Fig. 2). Depending on the level of confidence desired, a more lenient threshold can be chosen. Besides significance testing, $\alpha$ can be used for early stopping: for example, optimization can finish once a desired number of peaks in $\hat{\mathbf{x}}^{(k)}$ reach or exceed $\alpha$.

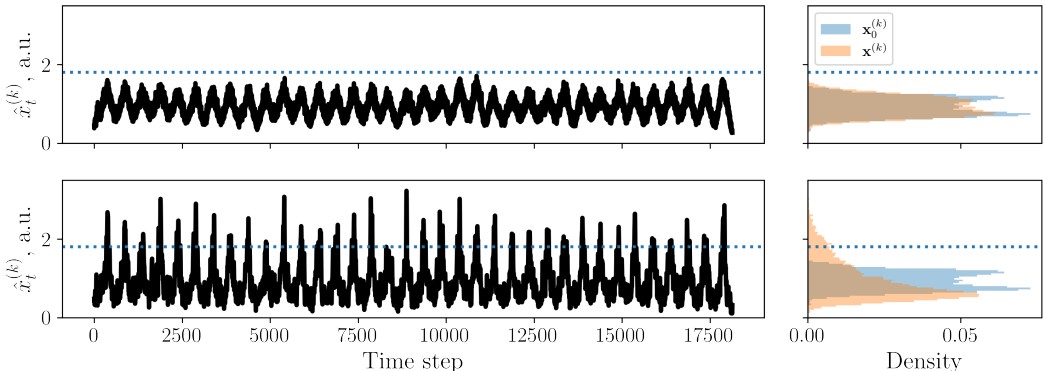

Figure 2: Before optimization (top row) no pattern is detected as the peaks in $\hat{\mathbf{x}}^{(k)}$ lie below $\alpha$ (dotted line). After optimization (bottom row) multiple occurrences of the pattern are detected. Red histograms in the panels on the right illustrate $\hat{\mathbf{x}}^{(k)}$ as densities before and after optimization compared to the density of values in $\hat{\mathbf{x}}_0^{(k)}$ expected from a random filter (blue histograms).

## 4 EXPERIMENTS

Since both formulations of our method perform comparably, here we report the results obtained using the first formulation.

### 4.1 ACCURACY PERFORMANCE METRICS

To evaluate the model's accuracy performance we use the following three metrics: *true positive rate* – the proportion of times a sequence is detected by its preferred filter. A detection is scored when the $k$-th filter responds with a significant peak in $\hat{\mathbf{x}}^{(k)}$ within no more than $M$ time steps of the ground truth label marking the middle of a sequence. This margin of $M$ time steps is needed because the response of an optimized filter to its preferred pattern is not guaranteed to coincide perfectly with the middle of the pattern. This is especially the case if a filter's chosen width exceeds the width of the pattern. *False positive rate* – the proportion of times a filter produces a significant peak to a non-preferred sequence or background activity (that is when no sequence is expressed). Finally, *false negative rate* – when a filter fails to produce a significant peak in response to its preferred sequence.

---

[1]Empirically, setting $\alpha$ to 4 standard deviations ensures a very low false positive rate.

## 4.2 SYNTHETIC DATA

We first test our method on three synthetic datasets. To simulate biologically realistic spike statistics, these datasets were constructed by embedding different spike sequences into a matrix of background activity $\mathbf{X} \in \{0,1\}^{N \times T}$ obtained by permuting the rows and columns of the real mouse CA1 recording described in Sec. 4.3. To facilitate comparisons, in all the experiments the shape of the synthetic datasets ($N = 452$, $T = 18137$) and filters ($N = 452$, $M = 200$) were kept the same unless indicated otherwise.

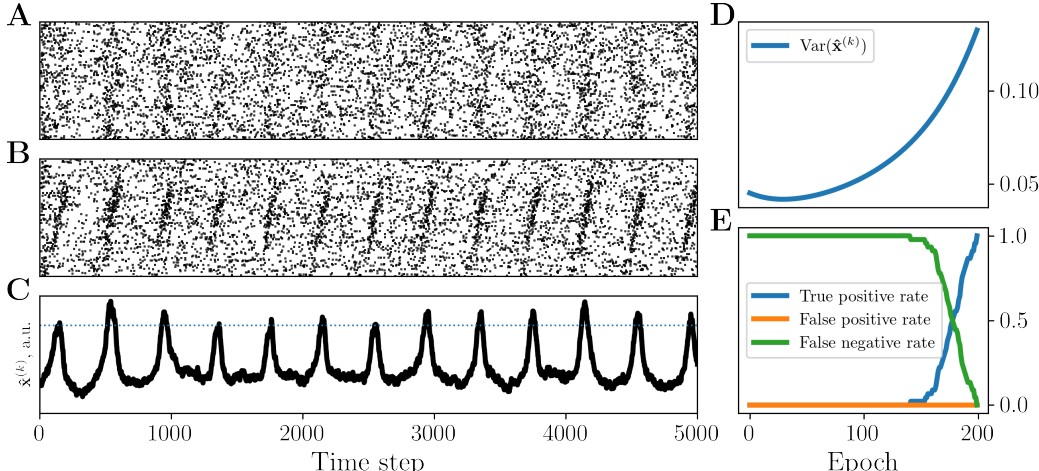

Figure 3: (A) and (B) depict the first 5000 time steps of the data before and after sorting based on the optimized filter, whose convolution with the data is shown in (C). (D) and (E) show the variance of the filter's response and performance metrics, respectively, until early termination.

**Experiment 1.** We first consider the simplest case, in which only one sequence is embedded. Each repetition of the sequence (45, 30 and 22 repetitions, 400, 600 and 800 time steps apart, respectively) consists of 80 neurons each of which is dropped with a probability of 0.2. We also add a Gaussian temporal jitter with a standard deviation of 10, 20 and 30 time steps. As illustrated in Fig. 3, the model is able to detect almost all the 45 sequence occurrences. Expectedly, the accuracy performance degrades as individual spike timings within a sequence occurrence deviate more from their ideal timing (higher spike jitter) and as the sequence occurrences become less frequent (longer inter-sequence interval). The accuracy performance also depends on how many spikes are dropped from the sequence (spike sequence sparsity), and the number of neurons participating in the sequence (sequence length). We provide additional test results and further details in Appendix B.

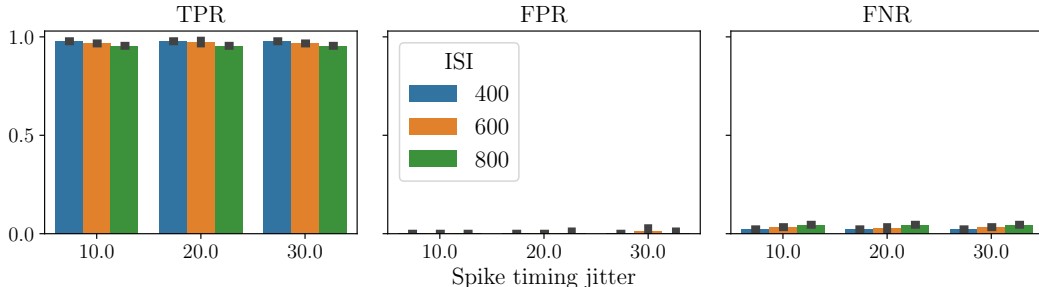

Figure 4: Model's accuracy performance as a function of spike timing jitter and inter-sequence interval (ISI). For additional experiments, see Appendix B.

Figure 4 provides insight about the limits of the model's ability to detect sequences: the false positive and false negative rate start to increase as the frequency (total number of sequence repetitions) decreases and spike timing jitter increases.

**Experiment 2.** We next test the ability of the model to detect two partially overlapping sequences. This is a more challenging scenario because the filters will have to compete for the neurons shared by both sequences. We used the same parameters as in Experiment 1, except that instead of 1 sequence of 80 neurons, we embedded 2 sequences of 250 neurons (overlapping by 50 neurons) alternating every 500 time steps. Overall, each sequence was repeated 18 times (36 repetitions in total).

Despite partial sequence overlap, the model is able to disentangle all the sequence occurrences correctly (Fig. 5). We note, however, the presence of undesirable peaks in the response of the second filter (Fig. 5C), which indicates that unidirectional patterns with shared neurons are hard to disentangle cleanly. Although those undesirable peaks do not reach the threshold of significance, they pose a potential issue for the detection of short or closely adjacent sequences with shared neurons. We leave detailed treatment of such cases as well as further improvements of the method to future work.

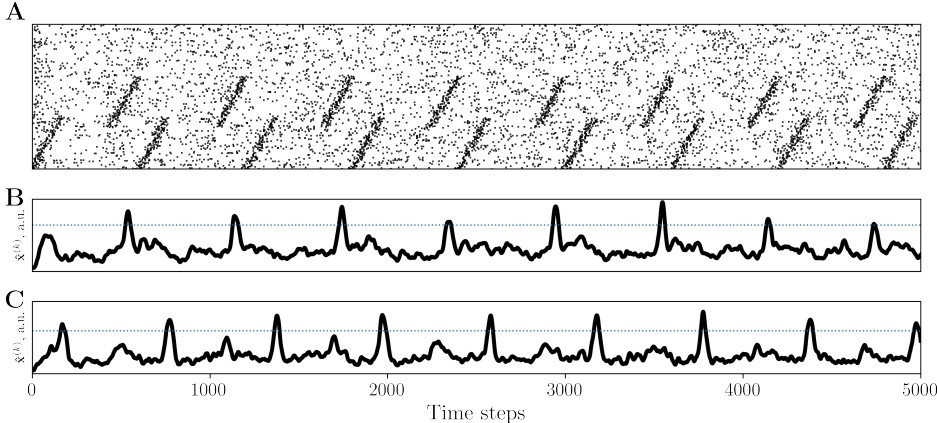

Figure 5: The model can correctly detect all the repetitions of two partially overlapping sequences. (A) fragment of the original data before permuting the rows. Response of the first and second filter after optimization are shown in (B) and (C), respectively.

**Experiment 3.** Our third synthetic dataset contained two *bidirectional* sequences (i.e. expressed in both forward and reverse order), constructed in the same way as in the previous experiment, but consisting of fully shared 100 neurons (Fig. 6).

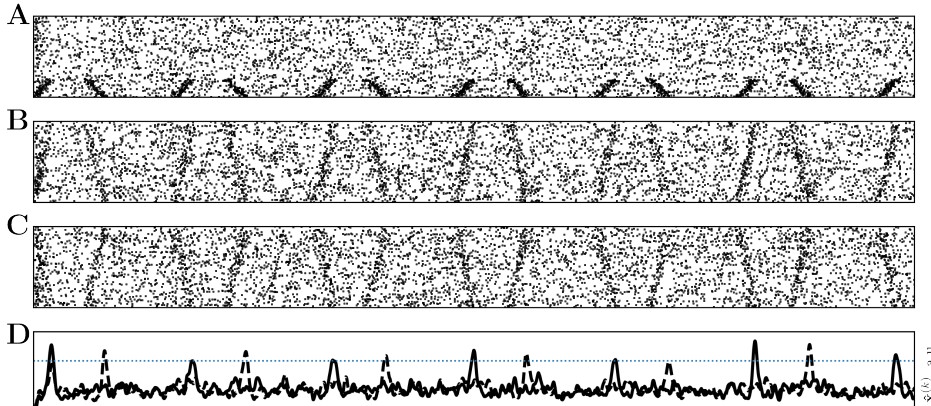

Figure 6: The model successfully detects forward and reverse instances of a bidirectional sequence. For illustration, the original data in (A) is shown before permuting the order of neurons. Sorting the original data with the first (B) and second (C) optimized filter exposes the forward and reverse sequences. The solid and dashed lines in (D) show the first and second filters' responses, respectively. The dotted horizontal line in (D) marks the significance threshold.

### 4.3 RECORDING FROM THE CA1 AREA OF THE MOUSE HIPPOCAMPUS

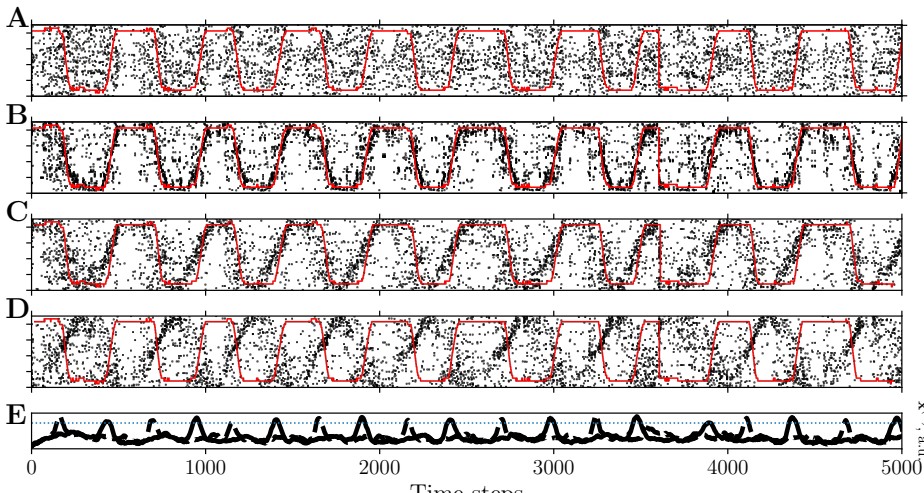

Figure 7: The red line in (A) and (B) indicates the animal's position on the track. In (B) neurons of the original dataset (A) are sorted according to their known place fields. In (C) and (D), neurons of the original dataset are rearranged with the indices obtained by sorting the first and second optimized filters, respectively. The first and second filters' responses are shown in (E) with solid and dashed lines, respectively. The dotted horizontal line in (E) marks the significance threshold.

Finally, we tested the ability of our method to expose place cell sequences in real neural data . We used a dataset [2] from Rubin et al. (2019), which is a recording of CA1 neurons of a mouse running on a linear track and collecting water rewards dispensed at its ends. In this experiment the position of the mouse was recorded simultaneously with the neural activity, and so we can verify the detected patterns against the ground truth – the ordering of neurons based on their known place fields. A neuron's place fields are determined by measuring its activity across the environment: the more a neuron fires in a particular location, the more "preferred" that location is. As the animal goes though different locations on the track, neurons with similar place tuning are more likely to spike together, and this information can be used to rearrange the order of neurons to make place cell sequences clearly visible (Fig. 7B).

With $K = 2$ and $M = 200$, our model was able to disentangle the forward and backward sequences of place cells, with peak activation of the preferred filters exceeding the significance threshold.

### 4.4 SPEED BENCHMARKS

We show how our method's run time scales as a function of dataset size compared to a selection of recently published pattern detection methods (*seqNMF*[3] and *PP-Seq*[4]). Using the same hardware, we ran the methods on a grid of datasets, each with the same number of neurons and sequence properties (Appendix D), but different number of timesteps, $T$, and intensity of background activity, $S$, defined here as $\frac{1}{NT} \sum \mathbf{X}$. Each optimization was run for 100 steps. 100 is the default number of optimization steps in the open-source implementations of *PP-seq* and *seqNMF*. In our model, the same number of optimization steps was sufficient for the $\mathrm{Var}(\hat{\mathbf{x}}^{(k)})$ term in the loss function to reach an approximate plateau, indicating no need for further optimization.

To ensure as fair a comparison as possible, we first run our method with GPU disabled (orange line in Fig. 8). Compared to *PP-Seq*, our approach is about 32 times faster on the largest dataset (500000 timesteps), and enabling the GPU further reduces the run time by a factor of six.

---

[2]The dataset is available at https://github.com/zivlab/island and represents a binary matrix obtained by thresholding the original $Ca^{2+}$ imaging data.

[3]https://github.com/FeeLab/seqNMF

[4]https://github.com/lindermanlab/PPSeq.jl

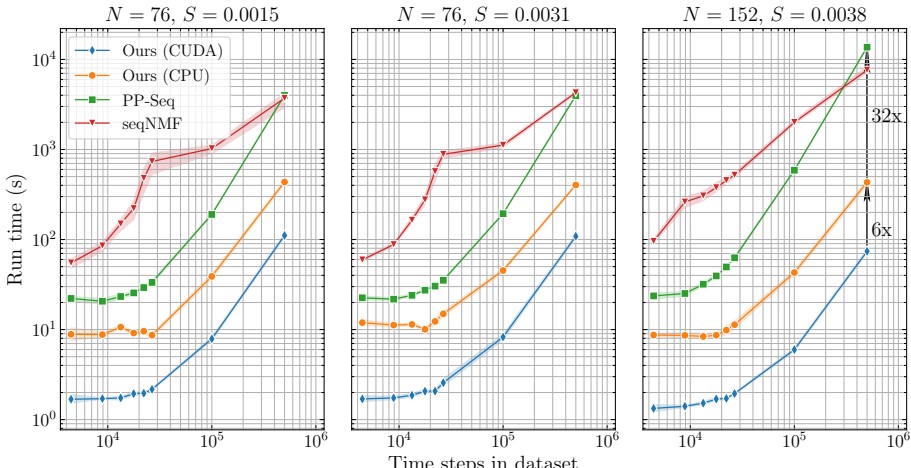

Figure 8: Regardless of the number of neurons ($N$) and intensity of the background activity ($S$), our method outperforms *seqNMF* and *PP-Seq* on the same datasets. Shades indicate 95% confidence intervals computed over 8 runs.

### 4.5 IMPLEMENTATION AND TRAINING NOTES

The model was implemented in Pytorch (Paszke et al., 2019) and optimized with the Adam (Kingma & Ba, 2014) optimizer with default parameters except the learning rate which was set to 0.1 for faster convergence. For the 2D convolution operation we used no padding in the dimension of neurons and a padding of $M//2$ zeros in the time dimension to ensure that $\hat{\mathbf{x}}^{(k)}$ has the same number of timesteps as the dataset. The cross-correlation term was implemented as 1D convolution with zero padding of size $M//2$. The total variation term smoothens the convolution $\hat{\mathbf{x}}^{(k)}$. We found it to be less important for the second formulation of our method, because the parameterization of $\mathbf{W}^{(k)}$ with truncated Gaussians itself has a strong smoothing effect on the corresponding $\hat{\mathbf{x}}^{(k)}$. In general, given the same dataset and filter sizes, one optimization step takes approximately the same time for both formulations of our method. All the experiments were run on a Linux machine with a 64-core AMD EPYC 7702 CPU with 503GB of RAM and an NVIDIA A6000 GPU with 48.67 GB of RAM. We use batch gradient descent, since all the datasets fit entirely into the RAM. However, implementing batched optimization (for even larger datasets or smaller RAM) is straightforward.

## 5 LIMITATIONS AND FUTURE DIRECTIONS

As with the other similar methods, one limitation of our proposed approach is the need to make assumptions about the number of sequences as well as their approximate duration. We considered relatively simple scenarios, in which the patters were similar to those observed in the hippocampi of rodents moving on a linear track, and the quality of the patterns detected can be verified by eye inspection. In other areas, patterns can be more highly variable or rare, making their detection more difficult (but possible, as we show in Appendix B and Fig. B.13). Testing the method's performance on more complex datasets, especially with weak and overlapping patterns, as well as exploring its possible extensions is an interesting direction for future work.

## 6 CONCLUSIONS

In this paper we have proposed a method for unsupervised detection of sequential patterns in neural recordings which may have practical utility in neuroscience research, especially in situations in which no behavioral references are available. We demonstrated that both on synthetic and real data, our approach is able to detect multiple spike sequences, including those that partially share neurons, or those that involve exactly the same neurons but are expressed in forward and reverse directions.

Importantly, our approach is much faster, which unlocks new possibilities for the study of structured spontaneous activity in large-scale neural recordings.

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

SUPPLEMENTARY MATERIAL

## A  HYPERPARAMETERS

Table 1: Hyperparameters

| Hyperparmeter | Description | Value |
|---|---|---|
| $\beta_{\text{xcor}}$ | Diversity loss weight | 0 if $K = 1$ else 10.0 ($10^5$ in Fig. B.12) |
| $\beta_{\text{TV}}$ | Total variation weight | 15.2 in Fig. 1, 4.5 in Fig. 8, otherwise 100.0 |
| $M$ | Filter width (along the time dimension) | 200 in Figs. 1 and 7, otherwise 100 |
| lrate | Learning rate | 0.1 |
| $j$ | Maximum cross-correlation distance | $M$ |
| $\sigma$ | Standard deviation of the filters' Gaussians (2nd formulation) | 16.0 (20.0 in Fig. B.12) |

In general, the total variation term can be set to zero in the formulation with truncated Gaussians (see main text), especially with relatively large values of $\sigma$. In cases that in involve overlapping sequences (as in Figs. 5, 6 and 7). We have also observed the need for a large weight for the cross-correlation penalty in Eq. 1

## B  SEQUENCE DETECTION PERFORMANCE

### B.1  DATA PREPARATION

To further evaluate how the model's accuracy performance depends on sequence properties, we constructed a grid, in which each dataset differed by the following four properties: (1) pattern sparsity (spike dropout probability), (2) inter-sequence interval (number of time steps between the sequences), (3) length (number of neurons in a sequence before applying dropout), and (4) jitter (standard deviation, in time steps, by which spike timing deviates from its ideal timing). Each dataset was constructed by embedding the sequences with a unique combination of these parameters into the same background activity matrix (452 neurons by 18137 time steps). The background activity matrix was obtained by permuting the rows and columns of the real recording of the CA1 area of the hippocampus of a mouse. On each of the datasets, the model was optimized for 3000 epochs (12 times to estimate the performance metrics' confidence intervals for each combination of sequence parameters). Figs. B.4, B.5, and B.3 suggest that the method performs best on sequences that are strong (i.e. involve relatively many neurons), dense (have a relatively low spike dropout probability), temporally stable (have a relatively low jitter) and well represented (occur relatively frequently).

325  B.2  OUR METHOD

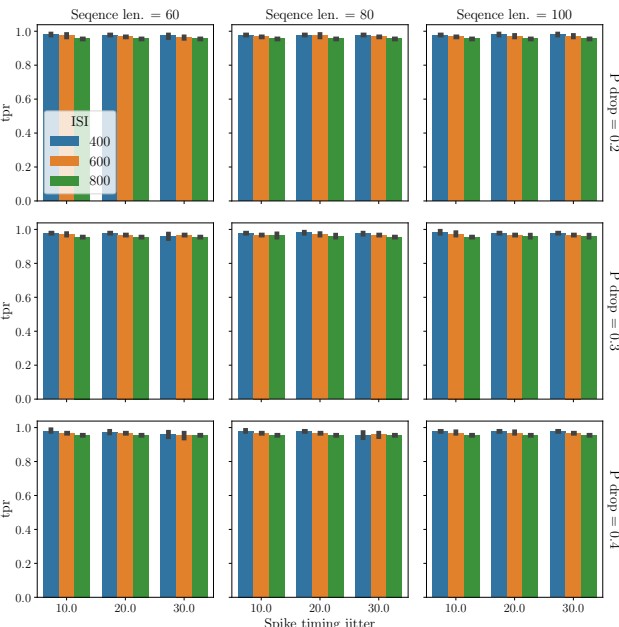

Figure B.1: Dependence of our method's TPR on sequence properties: sequence length, spike dropout probability, spike timing jitter and inter-sequence interval (ISI). Error bars are computed over 12 optimization runs (each with 3000 epochs).

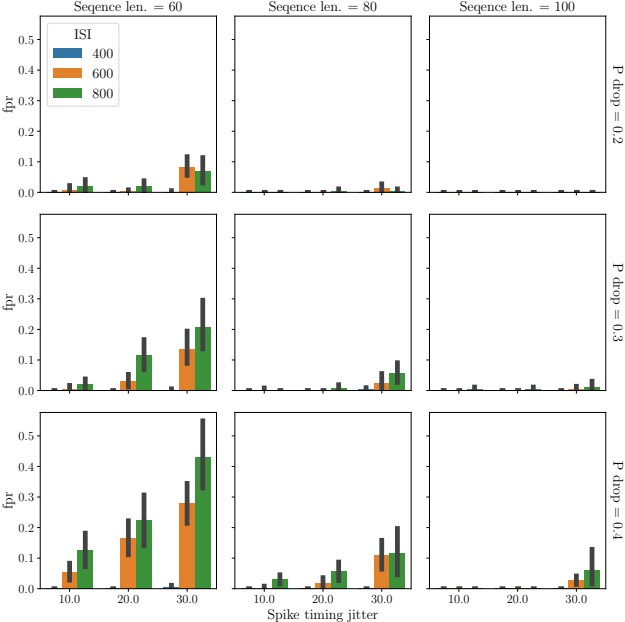

Figure B.2: Dependence of our method's FPR on sequence properties: sequence length, spike dropout probability, spike timing jitter and inter-sequence interval (ISI). Error bars are computed over 12 optimization runs (each with 3000 epochs).

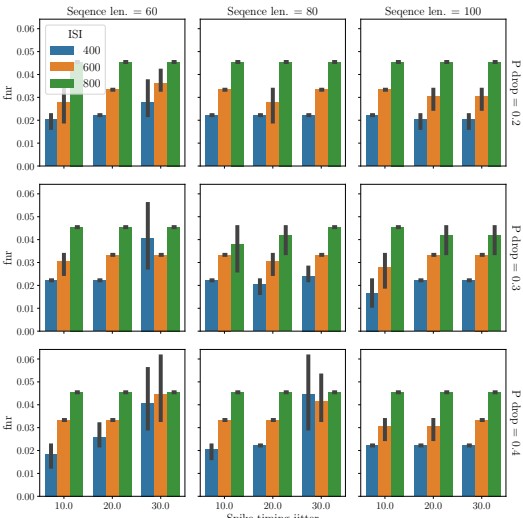

Figure B.3: Dependence of our method's FNR on sequence properties: sequence length, spike dropout probability, spike timing jitter and inter-sequence interval (ISI). Error bars are computed over 12 optimization runs (each with 3000 epochs).

## B.3  PP-SEQ

We also ran *PP-Seq* on the exactly the same grid of datasets with default parameters and found that that while its FNR was zero, the false positive rate (FPR) was higher than in our method. We used *PP-Seq* for comparisons because of its speed and because, unlike other similar approaches (e.g. *seqNMF*, *convNMF*), it explicitly outputs estimated times for pattern occurrences, making it easy to compute TPR, FPR, and FNR.

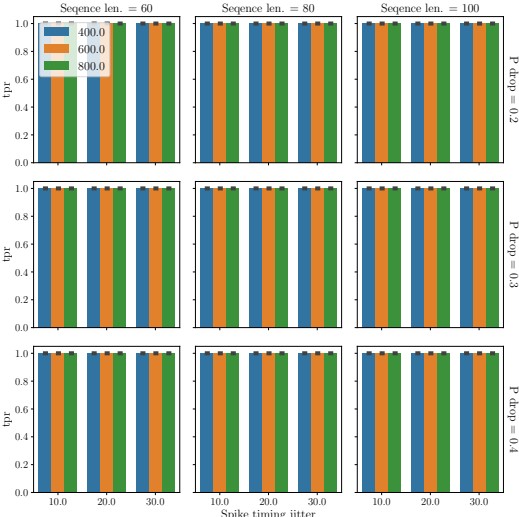

Figure B.4: Dependence of *PP-Seq*'s TPR on sequence properties: sequence length, spike dropout probability, spike timing jitter and inter-sequence interval (ISI). Error bars are computed over 12 optimization runs.

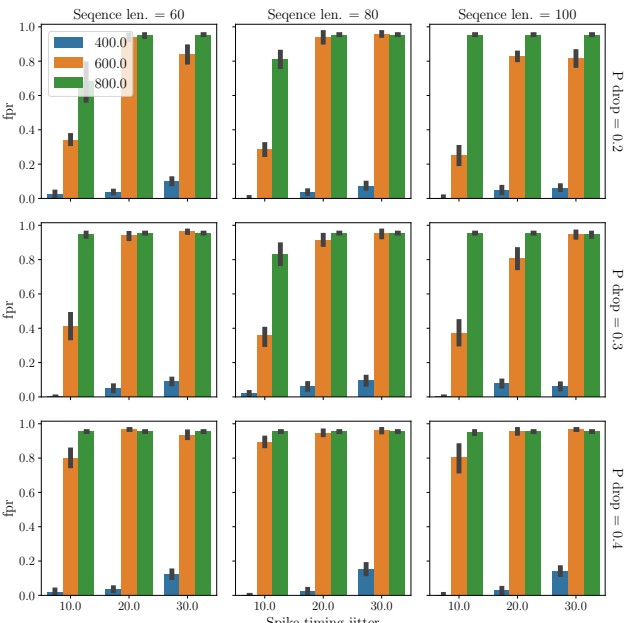

Figure B.5: Dependence of *PP-Seq*'s FPR on sequence properties: sequence length, spike dropout probability, spike timing jitter and inter-sequence interval (ISI). Error bars are computed over 12 optimization runs.

We do not show results for PP-Seq's FNR because of zero false negatives.

## C    ALGORITHMS

---
**Algorithm 1** With minimally constrained filters

---
**Input: X**, $K$, steps
**for** $k \in \{1, ..., K\}$ **do**
    Initialize $\mathbf{W}^{(k)}$
**end for**
**for** steps **do**
    Take a gradient step for $\mathcal{L}$
    Update $\mathbf{W}^{(k)}$, $k \in \{1, ..., K\}$
**end for**
**for** $k \in \{1, ..., K\}$ **do**
    Sort the rows of $\mathbf{W}^{(k)}$ according to the latency of the
    maximum within-row value, record sorting indices **s** of size $N$
    Obtain $\mathbf{X}^{(k)}$ by re-ordering the rows of **X** with **s**
**end for**

---

## D    DATASET AND SEQUENCE PROPERTIES USED FOR SPEED BENCHMARKS

Each dataset of $N \in \{76, 152\}$ neurons was constructed out of background activity matrices with $T \in \{4441, 8882, 13323, 17764, 22205, 26646, 100000, 500000\}$ timesteps and with background spiking intensity $S \in \{0.0015, 0.0031, 0.0038\}$. Into these background activity matrices we embedded sequences of 40 neurons, each with the following fixed parameters: dropout probability of 0.2, inter-sequence interval of 200 timesteps, and the standard deviation of spike timing (jitter) of 10 timesteps.

---

**Algorithm 2** With parameterized truncated Gaussians

---

**Input: X**, $K$, steps, $\sigma$
**for** $k \in \{1, ..., K\}$ **do**
   **for** $n \in \{1, ..., N\}$ **do**
      Make a truncated Gaussian $\mathbf{g}_n^{(k)}$ with mean $\mu_n^{(k)} \sim \mathcal{U}(1, M)$ and standard deviation $\sigma$
      Set the $n$-th row of $\mathbf{W}^{(k)}$ equal to $\mathbf{g}_n^{(k)}$
   **end for**
**end for**
**for** steps **do**
   Take a gradient step for $\mathcal{L}$
   Update $\mu_n^{(k)}$, $k \in \{1, ..., K\}$
   Construct a new $\mathbf{W}^{(k)}$, whose rows are truncated Gaussians with $\mu_n^{(k)}$, $k \in \{1, ..., K\}$
**end for**
**for** $k \in \{1, ..., K\}$ **do**
   Sort $\boldsymbol{\mu}^{(k)}$, record sorting indices $\mathbf{s}$
   Obtain $\mathbf{X}^{(k)}$ by re-ordering the rows of $\mathbf{X}$ with $\mathbf{s}$
**end for**

---

## E   TOTAL VARIATION

The total variation term encourages convergence to smooth $\hat{\mathbf{x}}^{(k)}$. We found that insufficient values of $\beta_{TV}$ increase the likelihood of a false positive (compare Fig. B.6 and Fig. B.7).

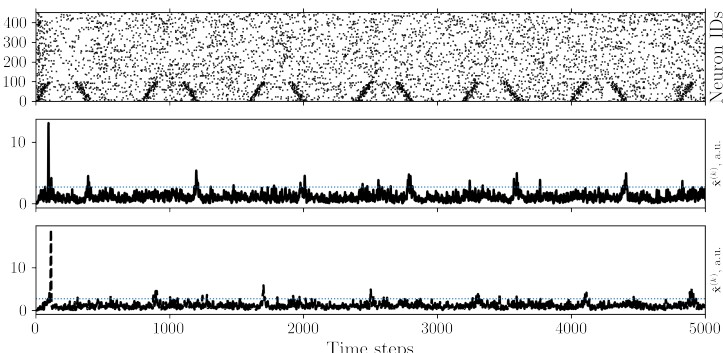

Figure B.6: With $\beta_{TV} = 1.5$, the model produces false positives. Middle and bottom panels show the response of the first and second filters, respectively.

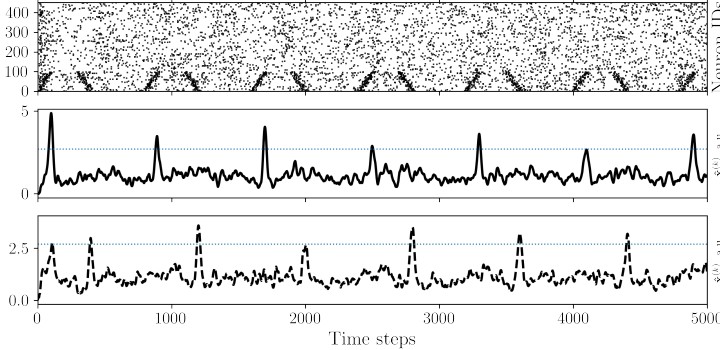

Figure B.7: With $\beta_{TV} = 100.5$, no false positives are present, the filters' responses (middle and bottom panels) are smooth and easy to interpret.

## F  RESULTS FOR THE SECOND FORMULATION OF THE METHOD

The results reported in the main text were generated using the first formulation of our method. To illustrate that the second method performs comparably, here we provide figures generated using the second formulation.

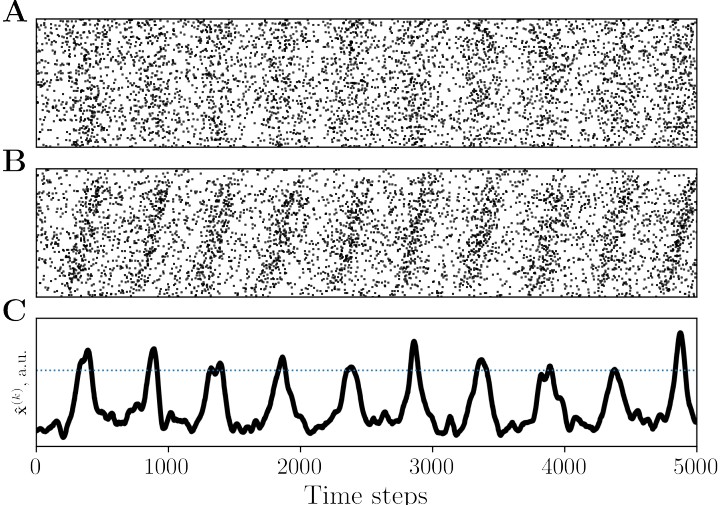

Figure B.8: Same as Fig. 1 in the main text.

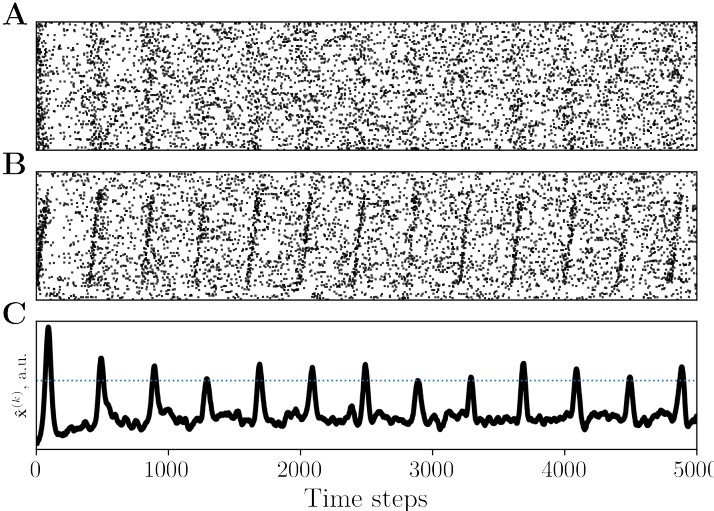

Figure B.9: Same as Fig. 3 in the main text.

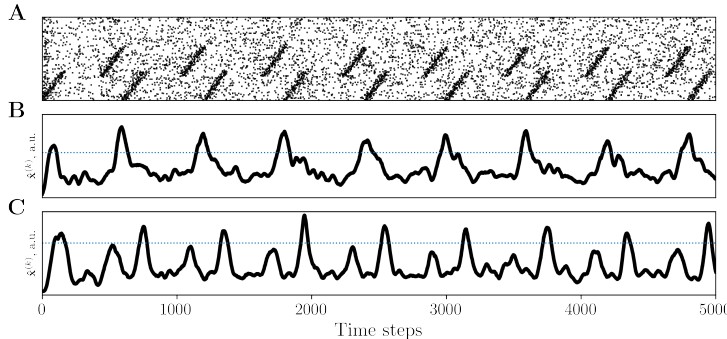

Figure B.10: Same as Fig. 5 in the main text.

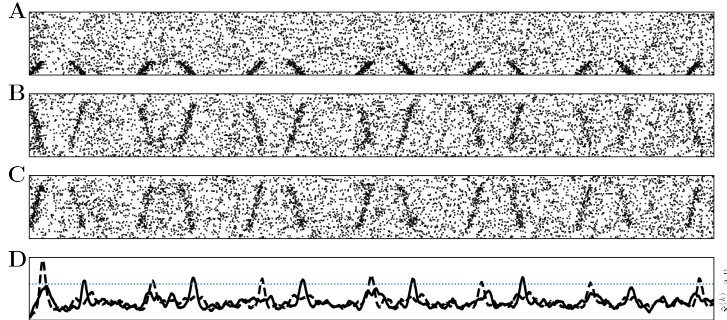

Figure B.11: Same as Fig. 6 in the main text.

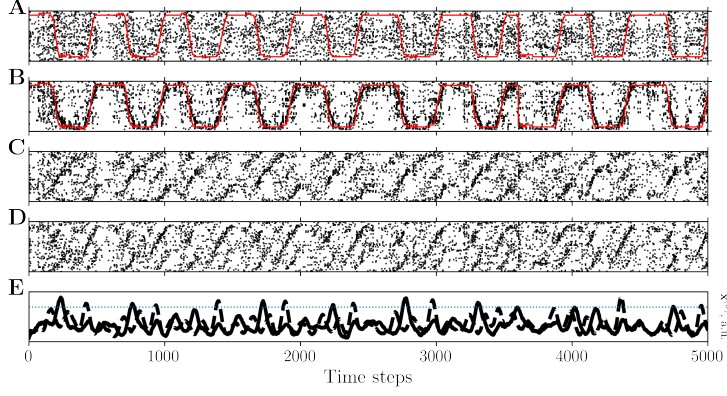

Figure B.12: Same as Fig. 7 in the main text.

## G   GUIDANCE ON CHOOSING $K$ AND $M$

**Choosing $K$.** Similarly to *seqNMF* and *convNMF*, our method still works if the number of filters $K$ is not exactly equal to the actual number of patterns. If $K$ is less than the number of patterns, the filters become tuned to the stronger of the patterns. In the reverse situation, when $K$ happens to be greater than the number of patterns, some "extra" filters' convolution curves will have a large degree of similarity, but their peaks will not reach statistical significance (e.g. as in Fig B.13 B and D). We found that a good strategy is to start with a conservative choice of $K$ (e. g. $K = 1$), and run optimization with progressively larger values of $K$ (such empirical search is realistic owing to the speed of our method). The significance of the convolution peaks provides a good guidance as to whether or not a particular choice of $K$ is good.

Consider the case in which two patterns exist in the data, and one them is much stronger than the other. With $K = 1$ the strongest of the two will be detected. With $K = 2$, both patterns will be detected (i.e. the first (already detected) one and the second). Setting $K = 3$ should result in still detecting the two patterns plus some spurious "pattern" by the third filter (whose convolution peaks should not reach statistical significance, because the cross-correlation term in Eq. 1 penalizes similar activations and, indirectly, filters that are tuned to the same pattern).

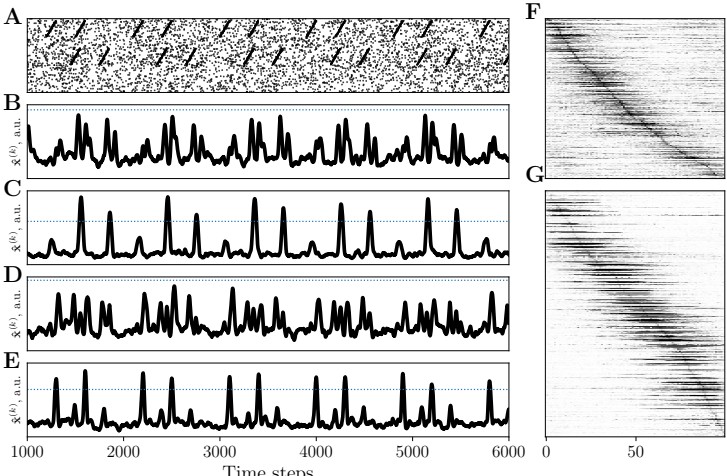

Figure B.13: Even with $K = 4$, which is greater than the number of sequences (2), and despite partial overlap in time the second and the fourth filters (C, E) recover the sequences. The extraneous two filters' peaks fail to reach the significance threshold (B, D).

**Choosing $M$.** As with $K$, $M$ should be chosen empirically, unless one has prior knowledge about the length of the expected sequences. It should be noted that

1. If $M$ is longer (more than twice the pattern's length) than the pattern, the share of the spikes participating in the pattern is too small relative to background spikes, effectively reducing the signal-to-noise ratio.

2. if $M$ is significantly shorter than the pattern (less than half the pattern's length), the filter might not "see" the pattern in its entirety, which might lead to more than one pattern being tuned to different parts of the same sequence (e.g. one tuned to the beginning and the other tuned to the end of the sequence).

## H    HANDLING TIME-WARPED SEQUENCES

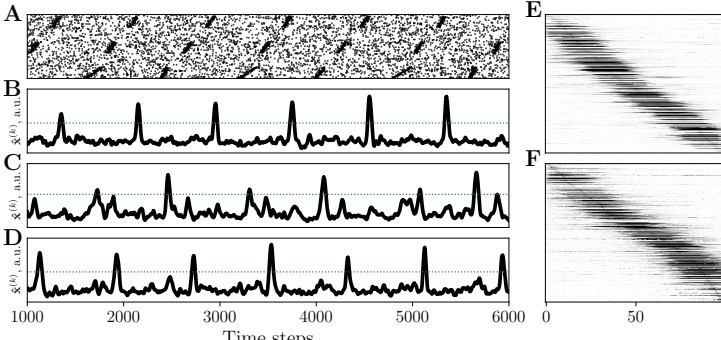

Figure B.14: The model detects 3 sequences one of which(bottom) is time-warped with a factor randomly chosen from $\{0.6, 1.0, 1.8, 2.2\}$. B, C and D show $\hat{\mathbf{x}}^{(k)}$, $k \in \{1, 2, 3\}$.

374  I   RUN TIME SCALING AS A FUNCTION OF $K$

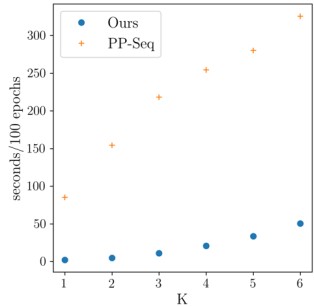

Figure B.15: Run time as a function of $K$ (sequence types). Dataset parameters: $T = 18137$, $N = 452$. One sequence of 40 neurons with dropout of 0.2, ISI of 200 time steps, and jitter of 10.

375  J   A CLOSER LOOK AT OPTIMIZED FILTERS

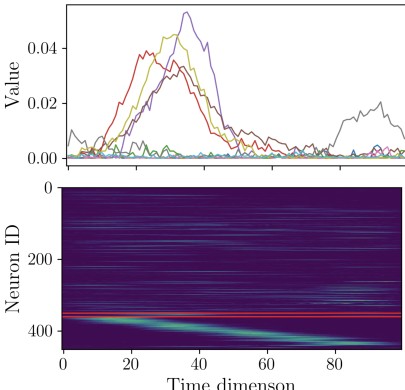

Figure B.16: Bottom panel: A sorted optimized filter. Upper panel: line plots of a selection of the filter's rows (delimited by the red lines). When a neuron is inactive in a pattern, its row in the corresponding filter appears flat, while for those that are active a Gaussian-like curve is observed.

