# OpenReview forum: "Unsupervised Detection of Recurrent Patterns in Neural Recordings with Constrained Filters"
_ICLR.cc/2024/Conference — Submitted to ICLR 2024_

### Official Review · Reviewer_C8pe · 2023-10-20

**Soundness:** 3 good
**Presentation:** 3 good
**Contribution:** 3 good
**Rating:** 6
**Confidence:** 3

**Summary:**

The authors propose a convolution dictionary learning method designed for neural data. They also propose a way to assess the statistical significance of their pattern detection and demonstrate speedup compared to other methods.

**Strengths:**

The paper is clearly written and there is an appreciable progression from experiments on synthetic data to real data. Moreover, the authors present a method to assess the statistical significance of their convolutional pattern detection.

Figures 3.D and 3.E are reassuring in that they seem to show that maximizing the variance in the objective eq.1 (which was motivated intuitively) does indeed correlate with pattern detection.

Finally, beyond the interpretability of their method, the authors exhibit a speedup compared to other methods.

**Weaknesses:**

I am surprised in Figure 8 that there are few standard convolutional dictionary learning methods to compare against, given that convolutional dictionary learning is a field with a rich literature. Could the authors explain how their method differs from other convolutional dictionary learning methods used for neural data, e.g. [1]?

[1]  Dupre La Tour et al. Multivariate Convolutional Sparse Coding for Electromagnetic Brain Signals. NeurIPS, 2018.

**Questions:**

Can the authors specify in the main text the data modality used in section 4.3.: are these measurements from cell calcium imaging?

Can the authors explain the main argument for the speed of their method compared to other methods in Figure 8?

What is f, on line 113?

---

> ### Author Response · Authors · 2023-11-15
>
> ***Could the authors explain how their method differs from other convolutional dictionary learning methods used for neural data, e.g. [1]?***
>
> Thank you for this question. convNMF, SeqNMF, convolutional sparse coding ([1]) aim to minimize the error (typically MSE) between the original signal and its reconstruction (obtained with some sparse filters and their activity, both of which are learned). By contrast, our method learns the filters only (or, in the second formulation of our method, only the means of per-neuron Gaussians, which are used to parametrize the filters).
>
> ***Can the authors specify in the main text the data modality used in section 4.3.: are these measurements from cell calcium imaging?***
>
> Yes. These are binarized calcium imaging data. We now make this explicit in the footnote on p. 7 (in addition to providing a reference to the original paper and a link from which the data was downloaded).
>
> ***Can the authors explain the main argument for the speed of their method compared to other methods in Figure 8?***
>
> We did not specifically check of PP-Seq or seqNMF for algorithmic inefficiencies. However, it seems that PP-seq is mostly bottlenecked by the expensive sampling operations during optimization. By contrast, our algorithm does not rely on sampling, and the compute-intensive matrix multiplications (which are at the core of our method) are very efficiently parallelized in PyTorch. For seqNMF, a possible bottleneck is the need to learn both the filters and their activations at the same time.
>
> ***What is $f$, on line 113?***
>
> $f(\cdot)$ is a truncated Gaussian function. We now make it explicit on line 111.

---

> > ### Comment · Reviewer_C8pe · 2023-11-21
> > **Acknowledging author response**
> >
> > I thank the authors for their response which addresses my concerns.

---

> > > ### Author Response · Authors · 2023-11-22
> > > **Thank you**
> > >
> > > We thank the Reviewer once again for the constructive feedback and the effort in reviewing our work.

---

### Official Review · Reviewer_u7XE · 2023-11-01

**Soundness:** 4 excellent
**Presentation:** 4 excellent
**Contribution:** 3 good
**Rating:** 8
**Confidence:** 3

**Summary:**

This paper presents a way to find repeating spike patterns in multi-channel neural data. The method uses a novel loss function to learn multiple kernels that respond strongly to different recurring patterns. It is tested and found to perform well on several synthetic datasets as well as a dataset of mouse place-cell responses for which the ground truth is known via the mouse’s position on a track. The method is shown to run more quickly than related past methods.

**Strengths:**

The paper addresses a substantial issue in analysis of large neural data. It seems to work well and efficiently. The method and the results are clearly presented.

The loss function seems elegant and well designed, and its explanation is clear. I didn’t find the loss obvious at first but rather felt that reading this section broadened my mind a little.

**Weaknesses:**

I take it the point of unsupervised detection of neural patterns is to find them even when ground truth isn’t known. However, the method wasn’t applied to such data. Such an application couldn’t be used to test the accuracy of the method, but it would help to illustrate qualitatively what can be expected from it in a realistic scenario, and it might provide an example of downstream use of the results.

**Questions:**

Line 74: Can “repeating” be defined more clearly? What kinds of variations aside from independent jitter are expected biologically, if any?

What are the spike rates of the background activity?

Figure 3E: Why does it appear that the network learns nothing for 150 epochs and then suddenly converges? This seems inconsistent with the choice of 100 steps in section 4.4, particularly the claim of faster convergence in lines 195-197.

Figure 7: Could the red traces be overlaid on panels C and D as well? Also it appears that the slopes in these panels are smaller than the speed of the mouse. Is that expected? Why? The detections are clear in any case, which is the main point.

Line 204: Is there really a 2D convolution operation? In the neuron dimension maybe you have a non-padded convolution with kernel size equal to input size, but I don’t think it’s standard to call that 2D.

Appendix B.4 & B.5: The comparison with PP-Seq is hard to interpret because both the true and false positive rates of PP-Seq are higher. Can you change a threshold to match one of these measures and compare the other?

The dropout probabilities range from 0.2 to 0.4, and I was not sure how to relate that to spike statistics (e.g. Poisson or otherwise). Can this be clarified?

Figure B.9: The sorted spike sequences look tighter here than in Figure 3. Are they? Why? Does it matter?

Figure B.12: This looks qualitatively quite different than Figure 7 and perhaps more should be said about this in the main text.

The learned kernels seem to include all the neurons, whether or not they participate in the sequence. Is it desirable to ignore non-participating neurons? Figure B.16 seems to suggest one way to do this, i.e. by checking for a Gaussian-like kernel. Are there better ways?

---

> ### Author Response · Authors · 2023-11-15
>
> ***I take it the point of unsupervised detection of neural patterns is to find them even when ground truth is known. However, the method wasn’t applied to such data. Such an application couldn’t be used to test the accuracy of the method, but it would help to illustrate qualitatively what can be expected from it in a realistic scenario, and it might provide an example of downstream use of the results.***
>
> That is correct. The point of unsupervised detection is to find patterns for which the ground truth is unknown. However, before we can trust any such method, it must be shown to produce results that make sense when checked against known ground truth. This is why we focused on cases with known ground truth.
>
> ***Line 74: Can “repeating” be defined more clearly? What kinds of variations aside from independent jitter are expected biologically, if any?***
>
> A “repeating” (which perhaps should be changed to “recurring”) pattern is one that appears more than a couple of times in the data. The occurrences of a pattern are not expected to be exact copies of each other (that is with same relative interspike intervals and number of participating neurons, in which case their detection would be much easier). Rather, the occurrences of a sequence are expected to differ due to spike timing jitter, dropout (that is when some neurons, otherwise part of a sequence, fail to fire) and noise (the presence of spikes from other neurons that are not part of the sequence).
>
> ***What are the spike rates of the background activity?***
>
> We assumed an average rate of 0.04 Hz based on the dataset from Rubin et al. (2019), which the original authors obtained by the thresholding Ca2+ imaging data of area CA1 of the hippocampus.
>
> ***Figure 3E: Why does it appear that the network learns nothing for 150 epochs and then suddenly converges?***
>
> This is because to be able to start detecting the sequences, the main objective (variance term in the loss function (Fig 3D)) needs to reach a certain level. In other words, as the filter optimization progresses, the filter becomes better “tuned” to the sequence, producing higher peaks when convolved with the data. Initially none of these peaks reach the statistical significance threshold (hence no true positives). At around 150 epochs a few peaks exceed the threshold and finally, at about 200 epochs, all the peaks reach significance.
>
> ***This seems inconsistent with the choice of 100 steps in section 4.4, particularly the claim of faster convergence in lines 195-197.***
>
> This is because of the different datasets: the one used in Fig 3E had 452 neurons and those used in the grid of runs in Section 4.4 had only 76 and 152 neurons. For those datasets, we observed that 100 epochs was sufficient for the $Var(x^{(k)})$ term in the loss function to reach a plateau (as we say on lines 195-197).
>
> ***Figure 7: Could the red traces be overlaid on panels C and D as well?***
>
> Yes, the traces are added in the updated manuscript.
>
> ***Also it appears that the slopes in these panels are smaller than the speed of the mouse. Is that expected? Why? The detections are clear in any case, which is the main point.***
>
> Indeed, the slopes do not perfectly match the slope of the sequence. This an expected artifact of rearranging the neurons based on optimized filters. Specifically, as explain in the Algorithms, in a given optimized filter, we find the positions of each row’s maxima and then sort the rows so that the maxima become temporally ordered. This may cause the recovered sequence to be stretched, vertically flipped or shifted in time (by a maximum of M/2 timesteps) compared to the ground truth. This is to be expected.

---

> > ### Author Response · Authors · 2023-11-15
> >
> > ***Line 204: Is there really a 2D convolution operation? In the neuron dimension maybe you have a non-padded convolution with kernel size equal to input size, but I don’t think it’s standard to call that 2D.***
> >
> > It is indeed a 2D convolution (we use the `torch.nn.Conv2D`). For each timestep, this convolution operation outputs one scalar. If we used 1D convolution, then we would have to use $N$ 1D kernels producing a vector of size $N$ at each timestep.
> >
> > ***Appendix B.4 & B.5: The comparison with PP-Seq is hard to interpret because both the true and false positive rates of PP-Seq are higher. Can you change a threshold to match one of these measures and compare the other?***
> >
> > In our algorithm, we set the significance threshold to four standard deviations (PP-Seq does not have a similar setting). Given that these figures were drawn based on a grid of simulations, scanning different thresholds would take a very long time. If the Reviewer insists, we will do so.
> >
> > ***The dropout probabilities range from 0.2 to 0.4, and I was not sure how to relate that to spike statistics (e.g. Poisson or otherwise).***
> >
> > Dropout was applied to vary the effective signal-to-noise ratio and test the method’s tolerance to noise. That is, the higher the dropout, the lower the SNR and the more difficult it is to detect a sequence. For simplicity, we did not specifically enforce any particular distribution of interspike intervals. However, to make the comparisons as fair as possible, all our synthetic datasets were constructed from the (permuted row- and columnwise) real CA1 dataset, which roughly preserves the original firing rates. This is mentioned on L144-147.
> >
> >
> > ***Figure B.9: The sorted spike sequences look tighter here than in Figure 3. Are they? Why? Does it matter?***
> >
> > Indeed, the sequences in B.9 look tighter than in Fig. 3, and this is this is expected. As we explained above, the raster rearranged with the filter(s) is not guaranteed to look identical to the ground truth due to sorting based on optimized filters. Also, while the results are mostly consistent from one optimization run to another, variations of the shape of the recovered sequences are expected unless the same seed for weight initialization is used. Finally, B.9 shows results for the second formulation of the method (in Fig. 3 we used the first formulation), and thus the difference is normal.
> >
> > ***Figure B.12: This looks qualitatively quite different than Figure 7 and perhaps more should be said about this in the main text.***
> >
> > The reason for the visibly imperfect recovery of sequences in B.12 is the same as we have explained in our answer to the previous question. If one is looking for repeated patterns, what matters ultimately is the presence of multiple significant peaks, while visual inspection of the rearranged neural data should be used as additional evidence in support of the presence of the peaks. We add a note about this on L103-105 of the updated manuscript.
> >
> > ***The learned kernels seem to include all the neurons, whether or not they participate in the sequence. Is it desirable to ignore non-participating neurons? Figure B.16 seems to suggest one way to do this, i.e. by checking for a Gaussian-like kernel. Are there better ways?***
> >
> > That is a great question. Indeed, ignoring non-participating neurons might (although we have not tried it) help produce visually cleaner patterns after rearranging the neurons with the optimized filters. One way to achieve this is to introduce $N$ learnable parameters constrained between 0 and 1, per each filter. These would act as per-neuron weights reflecting how actively the corresponding neuron participates in a sequence to which the filter is tuned (with 0 for no participation and 1 for maximum participation).

---

> > > ### Comment · Reviewer_u7XE · 2023-11-15
> > >
> > > Thanks again for your responses.
> > >
> > > Re. the convolution dimension: In a standard 2D image convolution, a 2D feature map is created by convolving a HxWxC input with a KxKxC kernel (C being the number of channels). Your description sounds analogous except with one fewer dimension. I guess you could implement it as Conv2D with one channel or Conv1D with multiple channels. But you have multiple channels of 1D data rather than one channel of 2D data, so the 1D terminology seems more suitable to me.
> > >
> > > Re. comparisons in B.4 and B.5, I don't see another way to achieve a clear comparison. I appreciate that a lot of computation is needed, but you might use a numerical method to quickly find the thresholds. Also, a less comprehensive grid of clearer comparisons might be more informative than the current comprehensive grid of hard-to-interpret comparisons.

---

> > > > ### Author Response · Authors · 2023-11-16
> > > > **Re: Conv2D, TRP/FPR comaprisons**
> > > >
> > > > ***Appendix B.4 & B.5: The comparison with PP-Seq is hard to interpret because both the true and false positive rates of PP-Seq are higher. Can you change a threshold to match one of these measures and compare the other?***
> > > >
> > > > We lowered the value of the significance threshold (down from 4 to 0.5 standard deviations) to roughly match the FPRs produced by PP-Seq and our method. This resulted in 100% TPR for our method (same as for PP-Seq). We include additional figures into the updated Supplementary Note (separate PDF document).
> > > >
> > > > ***Re. the convolution dimension: In a standard 2D image convolution, a 2D feature map is created by convolving a HxWxC input with a KxKxC kernel (C being the number of channels). Your description sounds analogous except with one fewer dimension. I guess you could implement it as Conv2D with one channel or Conv1D with multiple channels. But you have multiple channels of 1D data rather than one channel of 2D data, so the 1D terminology seems more suitable to me.***
> > > >
> > > > That is correct, we use Conv2D with one channel, and neurons are not treated as channels. To be specific, assuming $K = 2$, the matrices that go into the Conv2D operation have the following shapes:
> > > >
> > > > ```python
> > > > import torch.nn.functional as F
> > > > out = F.conv2d(
> > > >       X,                                 # data matrix of size [1, 1, N, T]
> > > >       weight=W,                  # filter matrix of size [2, 1, N, M]
> > > >       padding=(0, M//2),  #  no padding for the neurons’ dim, half the filters’ width for time
> > > >       bias=None)
> > > > ```
> > > >
> > > > `out` has shape `[1, 2, 1, T]` which, after squeezing the singleton dimensions, becomes `[2, T]`. These are the convolution curves $\mathbf{\hat{x}}^{(0)}$ and $\mathbf{\hat{x}}^{(1)}$.

---

> > ### Comment · Reviewer_u7XE · 2023-11-15
> >
> > Thank you for the clarifications. My question about slopes was unclear. I take it a slope could stretch or contract due to non-uniform density of place fields of neurons in the dataset. However, shouldn't the whole sequence be contained within the movement time? I don't see why an ordered sequence of spikes would take longer than the associated movement. If this is expected, maybe an explanation could be added to the caption or methods.

---

> ### Author Response · Authors · 2023-11-16
> **Re: slopes**
>
> ***Thank you for the clarifications. My question about slopes was unclear. I take it a slope could stretch or contract due to non-uniform density of place fields of neurons in the dataset. However, shouldn't the whole sequence be contained within the movement time? I don't see why an ordered sequence of spikes would take longer than the associated movement. If this is expected, maybe an explanation could be added to the caption or methods.***
>
> Thank you for the chance to clarify this further. After rearranging the neurons with an optimized filter a sequence can get stretched up to the temporal width of the filter (which was 200 steps). To illustrate this in Fig S7 (Supplementary Note), we overlay a transparent box spanning 200 time steps over the same raster as in Fig. 7. We agree that the appearance of the recovered sequence (in Fig 7 C and D) suggests that the animal was moving closer to the ends of the track (while it was actually stationary). This is an artifact of sorting the data with imperfectly optimal filters.

---

> > ### Comment · Reviewer_u7XE · 2023-11-17
> >
> > Sorry, I am still missing something and maybe I have misunderstood what is actually being plotted. If I understand correctly the plots are calcium-event rasters, and all you have done to them is re-order the rows. Of course the method finds repeating event patterns that are within the temporal width of the filter. However, the method doesn't shift the events themselves in time, does it? Does the plot not show an orderly progression of events that lasts longer than the movement?

---

> ### Author Response · Authors · 2023-11-17
>
> Your understanding is correct: we only rearrange the order of the rows and do nothing else. The individual spikes are never shifted in time. We admit that, perhaps due to some sort of visual illusion, sequences might appear longer than they actually are. To completely clarify the sorting procedure, we attach as Supplementary Material minimal code and necessary data to reproduce the entire process of sorting the rows of the raster to reveal the sequences. It is quite short an we hope this will make this matter clear.

---

### Official Review · Reviewer_qa1s · 2023-11-01

**Soundness:** 2 fair
**Presentation:** 3 good
**Contribution:** 2 fair
**Rating:** 5
**Confidence:** 3

**Summary:**

This paper presents an unsupervised method for identifying sequences in neural spiking data.   The method learns a set of K filters that summarize the spiking data subject to what seem like some pretty minimal constraints.  The method is applied to ground truth data and recordings from rodent hippocampus.  The method is faster and perhaps more reliable than other competing methods.

**Strengths:**

Right now, the priors that working neuroscientists bring to analyzing their data has a huge effect on the results they are able to discover.  It would be really very useful to have an unsupervised method to automatically extract sequential information from spiking neurons.  Not only would it be outstanding for analyzing data from freely moving animals (as in Fig 7)  but also would be really impactful for understanding population burst events, theta sequences, etc etc.

Such tools will become increasingly important as recording techniques continue to advance.

**Weaknesses:**

I am concerned about priors that may be ``baked in'' to the method (perhaps inadvertently).  At mimimum these priors should be made more explicit.  In particular, I'm concerned that the model seems to find ``straighter'' sequences than are present in the data (Fig 7).  The ground truth experiments all use linear sequences, exacerbating this concern.

It's not obvious that the method can generalize to sequences (such as PBEs, theta etc) that unfold over more than 1 continuous dimension.

**Questions:**

If ground truth includes sequences that unfold at varying rates can this method identify them?  For instance, suppose that there are place fields along a linear track but they are overrepresented near the ends.   The animal runs at a constant velocity. Now the sequence, rather than appearing as a straight line in, say, Fig 3b, would appear as a hook.  Can this method find those sequences as well?  I think this is a very important question as it seems that these kinds of sequences are very general.  Is there a way to make the filters more or less sensitive to these kinds of sequences?

Suppose we had a set of place cells that tile a 2-D enclosure.  Would this method work?  I'm concerned that the filters will have to cover a 2-D surface with piecewise 1-D filters and this will fail really badly.

Take the situation in Fig. 7.  Suppose the animal starts out on the linear track at a constant velocity, stops half way through, backtracks for 10 cm, then turns around and continues along its original trajectory to the end.  What filters does this method find?  What should it identify?

---

> ### Author Response · Authors · 2023-11-15
>
> ***I am concerned about priors that may be “baked in” to the method (perhaps inadvertently). At mimimum these priors should be made more explicit. In particular, I'm concerned that the model seems to find straighter' sequences than ¬are present in the data (Fig 7).***
>
> The sequences in Fig. 7 (panels C and D) show the spike raster rearranged with one of the two optimized filters. The objective only maximizes the detection strength, but imposes no prior on how the rearranged sequences will look in the raster. The seeming straightness or the sequences is an artifact of this (imperfect) rearrangement, not the effect of priors. To illustrate this, compare Fig. B. 12 and Fig. 7 (in the main text). Even though the datasets were exactly the same, the sequences look somewhat different. Unless the same seed is used to initialize all the trainable parameters of the model, the look and shape of the recovered sequences will differ from run to run. Most importantly, the main purpose of rearranging the rows of the spike raster is to help the researcher to visually confirm the presence of a sequence detected by convolution peaks exceeding the significance threshold.
>
> ***The ground truth experiments all use linear sequences, exacerbating this concern.***
>
> We assume that Reviewer refers to a "linear sequence” as a pattern where neurons (or ensembles of neurons) A, B, C, and D are activated in a strict predictable order (e. g. A → B → C → D, thus following a linear path). In that case, a "non-linear sequence” would involve a more complex pattern (e. g. a branching sequence like A → B → C or D). In fact, our approach detects such “non-linear” sequences without a problem if they fit entirely within the width of the filter ($M$). We illustrate this in a Supplementary Note, which we upload as Supplementary Material.
>
> ***It's not obvious that the method can generalize to sequences that unfold over more than 1 continuous dimension.***
>
> Thank you for the chance to clarify this. The method works for any pattern if it fits entirely within the filter’s temporal width ($M$). To substantiate this, in the Supplementary Note (that we have uploaded as a separate document) we consider an artificial 2D case in which, place cells tile a 2D enclosure, and this enclosure is traversed by an animal along two trajectories resulting in two sequences that unfold over more than 1 continuous dimension. The place cell neighboring in 2D space often end up far apart in the 1D space of the raster. However, even though this “locality” is broken, the method still works fine.
>
> ***If ground truth includes sequences that unfold at varying rates can this method identify them?***
>
> Yes. In fact, the method is tolerant to moderate time-warping of sequences (we illustrate this in Appendix H).
>
> ***… suppose that there are place fields along a linear track but they are overrepresented near the ends. The animal runs at a constant velocity. Now the sequence, rather than appearing as a straight line in, say, Fig 3b, would appear as a hook. Can this method find those sequences as well? I think this is a very important question as it seems that these kinds of sequences are very general.***
>
> Yes. We demonstrate this in Section 4.3.  Fig. 7. shows how our method identifies the sequences of places cells of a mouse running on a linear track. If we look closely at the sorted raster (panels C and D), we can see that the straight line of the sequences bends at the ends slightly, forming what looks like a “hook” (e.g. around 800 and 1000 timesteps). We can’t be sure though if this is due to the overrepresentation of track ends by place fields or due to the animal making brief stops. Importantly, if one is looking for repeated patterns, what matters ultimately is the presence of multiple significant peaks, while visual inspection of the rearranged neural data can be used as additional evidence for the presence of patterns.
>
> ***Is there a way to make the filters more or less sensitive to these kinds of sequences?***
>
> Thank you for this important question. If the researcher has strong assumptions (or knowledge) about what kinds of sequences they are looking for, it is possible in principle to incorporate these assumptions (or inductive biases) through the use of additional learnable parameters and regularization terms in the loss function.
>
> ***Suppose we had a set of place cells that tile a 2-D enclosure. Would this method work? I'm concerned that the filters will have to cover a 2-D surface with piecewise 1-D filters and this will fail really badly***
>
> On the contrary, our model uses 2D, not 1D, filters (as we explain on L72). This allows the detection of any (sufficiently strong) pattern as long as it fits entirely within the temporal width of the filter ($M$). We illustrate a 2D case in the Supplementary Note uploaded with the revised manuscript.

---

> > ### Author Response · Authors · 2023-11-15
> >
> > ***Take the situation in Fig. 7. Suppose the animal starts out on the linear track at a constant velocity, stops half way through, backtracks for 10 cm, then turns around and continues along its original trajectory to the end. What filters does this method find? What should it identify?***
> >
> > Thank you for the interesting question. If we assume that the animal runs back and forth most of the time and only backtracks once or twice, then the model is unlikely to learn a separate filter to capture this rare backtracking event, but will only learn two filters – one corresponding to the animal running in one direction and the other to it running in the other direction. Conversely, if the animal backtracks often, then the model is likely to learn a separate filter tuned to capture this zig-zag pattern. Finally, if the animal backtracks all (or almost all) the time, then the model will learn two filters (again, one for the forward direction and the other for the return trip).

---

> > > ### Comment · Reviewer_qa1s · 2023-11-22
> > > **Response**
> > >
> > > Thanks to the authors for their careful reply.
> > >
> > > I remain concerned that there is a scale built in to the method.  Choosing the width M may impose a choice of the experimenter on the results that are not present in the data.
> > >
> > > Hippocampal neurons (and neurons in other parts of the brain) appear to show reliable sequential firing over many different time scales.  For instance, the ``sequence'' of time cells triggered by an event slows continuously and may extend out to minutes (Shikano, et al., 2021, Current Biol).  The time between the peak of time cell n and the peak of time cell n+1 changes systematically with the location in the sequence.  (There is reason to suspect that place cells show a similar phenomenon.) If Cao et al., (2022, eLife) are to be believed, the time between cells in the sequence goes up linearly with n.   If the sensitivity of the method depends on the choice M, then any specific window size means that the method would be blind to parts of the sequence slower than that (and perhaps has different resolution for parts of the sequence that are much faster).
> > >
> > > I'm supportive of the general approach.  I encourage more thought on sensitivity to M, especially in the case where the data has sequences that unfold over many different characteristic scales.

---

> ### Author Response · Authors · 2023-11-23
>
> ***I remain concerned that there is a scale built in to the method. Choosing the width M may impose a choice of the experimenter on the results that are not present in the data.***
>
> Thank you very much for the opportunity to clarify this point. We would like to stress that if no pattern is present in the data, then our method will not detect anything (spurious regularities in the background neural activity should not produce peaks exceeding statistical significance). Regarding the choice of $M$, indeed, as we admit in the Limitations section, one must make some sensible assumption about the length of patterns that they are looking for in the dataset. This not problematic at all, because the speed of our method allows one to test multiple different values of $M$ in a reasonable amount of time.
>
> ***Hippocampal neurons (and neurons in other parts of the brain) appear to show reliable sequential firing over many different time scales <...> if the sensitivity of the method depends on the choice M, then any specific window size means that the method would be blind to parts of the sequence slower than that (and perhaps has different resolution for parts of the sequence that are much faster).***
>
> It is correct that if a pattern extends far beyond the filter’s temporal width, it will not be detected. However, it is not difficult for one to look for patterns at more than one time scale. For example, if a researcher hypothesizes that a certain behavioral task (or experimental treatment) causes an elongation of pre-existing patterns of neural activity, they can test this hypothesis by running the method first with a narrow window (e.g. $M$) on a dataset recorded before the animal performs the task and once again with a wider window (e.g. $2M$) on a dataset recorded after the task.

---

> ### Author Response · Authors · 2023-11-23
>
> To further illustrate our point we have added a case study to the most recent revision of the Supplementary Note (separate PDF file). Specifically, we consider a synthetic dataset ($N = 452$) with one sequence of 160 neurons that unfolds at different speeds (the second is 3 times slower than the first). We begin by setting $M = 200$ and optimize the first filter. As expected, the optimized filter only “responds” to the short version of the sequence (Fig. S8), because the longer sequence is not fully contained within the filter’s temporal length. Next, we set $M = 600$ and optimize the second filter. After optimization, this wider filter produces higher peaks in response to the longer sequence (Fig. S9). In general, the patterns do not need to be straight (or linear) -- the method works with any pattern configuration as long as it fits within $M$.

---

### Author Response · Authors · 2023-11-15
**Thank you to all the Reviewers**

We sincerely thank all the Reviewers for their insightful comments and suggestions, as well as the time they took to review our paper. We have worked hard to address all the concerns and questions in as much detail as possible and sincerely hope that the Reviewers will find our responses satisfactory. In addition to our responses, we uploaded a Supplementary Note (as a separate PDF document), and made changes to the manuscript (we specify the places in which the changes were made).

---

### Meta-Review · Area_Chair_yNqG · 2023-12-15

**Metareview:**

This paper proposes a back-propogation approach to learning spatio-temporal filters that extract sequences of neural activity from high dimensional time-series. The method improves timing-wise on past methods, such as seq-NMF and conv-NMF (non-negative factorization approaches), and provides statistical significance measures as well. It specifically differs from past factorization methods by only learning the filter. The authors demonstrate their method on a number of datasets including real neural datasets.

While there were a number of positive remarks, including on the importance of the problem and demonstrated results, there were a few points that came up during the review. With respect to the data, the use of binarized calcium imaging data is an odd choice as the height of the peaks carries a significant amount of information related to the short-term firing rate, especially in the burst-like activity in hippocampus. With respect to benchmarks, the only benchmarks were run for the speed tests, which does not show if the proposed change in cost function improves significantly on, e.g., the ability to identify sequences under different conditions. Finally, there was some confusion among the reviewers about the model and the place of convolution. While the latter point I believe was addressed eventually in the response, the former two points seem to limit the overall impact of this work in it's current form.  Therefor as this is a borderline case I am recommending the work not be accepted.

**Justification For Why Not Higher Score:**

The primary reasons for my decision are the lack of benchmarks, data processing choices, and model clarity (see main review).

**Justification For Why Not Lower Score:**

N/A

---

### Decision · Program_Chairs · 2024-01-16

Reject